# WL-Tree: a New Tool for Analyzing Graph Neural Networks

## Abstract

The 1-WL algorithm provides a clean algorithmic model for graph neural networks (GNNs) that run with a message-passing architecture. Previous work compares a GNN against the 1-WL algorithm to analyze its expressiveness and then develops new GNN variants under the guidance of the comparison. In this work, we propose WL-Trees, a new algorithmic model of GNNs. We compute WL-trees using a variant of Breadth-First-Searches on the input graph. We show that WL-trees are equivalent to colors computed from the 1-WL algorithm. Despite the equivalence, WL-trees deepen the understanding of a graph's structural information encoded in node representations. They also serve as an algorithmic model for improved GNNs to analyze their expressiveness from a new angle.

## 1 Introduction

Graph neural networks (GNNs) (Wu et al., 2020) have gained success in a series of graph learning tasks including node label predictions (You et al., 2018), link predictions (Zhang & Chen, 2018), graph classification (Errica et al., 2019), and graph generation (Li et al., 2018; You et al., 2018). Learning node representations is an indispensable step in all these graph learning tasks (Wu et al., 2020). An important question is: what structural information is encoded in node representations?

A fundamental form of GNNs is the Message-Passing Neural Network (MPNN) (Gilmer et al., 2017), which uses neural layers to compute messages and then passes them between neighboring nodes. At the output layer of an MPNN, each node gets a representation that encodes the node's structural information (Srinivasan & Ribeiro, 2019; Donnat et al., 2018). Two nodes with similar surrounding graph structures, which include node attributes in the range, should have similar representations, irrespective of their distance in the graph. More recent research shows that such node representations can be enhanced by positional information (Li et al., 2020; Dwivedi et al., 2021; Lim et al., 2022), which helps to model "spatial similarity" between nodes.

The 1-WL algorithm (Grohe et al., 2017; Grohe, 2017) is an algorithmic model of the MPNN given that they have the same message-passing structure. So theoretical analysis (Xu et al., 2018; Chen et al., 2020; Jegelka, 2022; Sato, 2020) uses the 1-WL algorithm to analyze the expressiveness of MPNNs. The main result is that an MPNN is no stronger than the 1-WL algorithm: if two graph nodes obtain the same color from the 1-WL algorithm, then they will get the same representation from the MPNN calculation. Guided by the analysis, new GNNs are designed to either match (Xu et al., 2018) or exceed the expressiveness of the 1-WL algorithm (Morris et al., 2019; Li et al., 2020; Chen et al., 2020; Sato et al., 2021; Zhang & Li, 2021; Dasoulas et al., 2021; Balcilar et al., 2021; Bouritsas et al., 2022).

The graded modal logic (GC2) (Barceló et al., 2020; Grohe, 2021) provides another equivalent algorithmic model for MPNNs. This is because logical tests in GC2 are equivalent to the 1-WL algorithm (Cai et al., 1992). Therefore, the expressiveness of MPNN can also be characterized by logical tests in GC2. The analysis of GNNs with different algorithmic models gives understanding of their expressiveness from different angles.

In this work, we propose the *WL-tree*, which is another algorithmic model for MPNNs. In particular, we construct a WL-tree from a Breadth-First-Search (BFS) that allows revisits. We show that WL-trees are equivalent to node colors computed by the 1-WL algorithm. Despite its equivalency with the 1-WL algorithm, WL-trees provide a more intuitive understanding of graph structures that can

be possibly learned by an MPNN. In this work, we focus on node representations. Our analysis with WL-trees has reached a new understanding of node representations. We provide an algorithm to identify all graph structures surrounding a node that give the same node representation. Our analysis also shows the usefulness of node representations in subgraph matching models. Finally, we use WL-trees to analyze how two GNNs enhance MPNN's abilities through richer node inputs.

In summary, WL-trees provide a new tool for the analysis of GNNs. Given their close relationship with the original graph, WL-trees enable a more intuitive and insightful understanding of node representations, which suggests new directions of improving GNNs.

## 2 RELATED WORK

A series of expressiveness analyses compare GNNs against the 1-WL algorithm in terms of distinguishing graph structures (Sato, 2020; Grohe, 2021; Jegelka, 2022). This algorithm is further used to direct the development of new GNNs. Xu et al. (2018) propose the Graph Isomorphism Network (GIN) to achieve the same level of expressiveness as the 1-WL algorithm. After that, a series work has been proposed to further improve GNN's expressiveness (Morris et al., 2019; Li et al., 2020; Chen et al., 2020; Sato et al., 2021; Zhang & Li, 2021; Dasoulas et al., 2021; Balcilar et al., 2021; Bouritsas et al., 2022).

Barceló et al. (2020) and Grohe (2021) use logic to characterize the expressiveness of GNNs. In particular, they show that the expressiveness of MPNNs is bounded by by GC2 tests, which is equivalent to the 1-WL algorithm (Cai et al., 1992). Guided by the FOC2 logic, Barceló et al. (2020) has designed a new GNN, whose improved expressiveness is equivalent to FOC2 logic tests. This work aims to develop another algorithmic tool for the analysis of GNNs.

Previous work also use trees to represent graph structures that can be described by the 1-WL algorithm. For example, Shervashidze et al. (2011) use the 1-WL algorithm to construct trees to compute graph similarity. Zhang & Li (2021) consider node representations with trees. However, these trees obtained by rolling out message-passing steps are different from our WL-trees: the latter are derived from from BFSes on graphs. Furthermore, there is no systematic investigation of the relationship between trees and GNN representations.

## 3 BACKGROUND

Let $G$ denote a graph with node set $V(G)$ and edge set $E(G)$. In this work, we only consider connected simple graphs. Let $\mathcal{N}(i) = \{j : (i, j) \in E(G)\}$ denote the neighbor set of a node $i \in V(G)$. Let $\text{dist}(i, j)$ denote the distance between two vertices $i, j \in V(G)$. If $i = j$, then $\text{dist}(i, j) = 0$. Assume each node $i$ is associated with a color $c_i$. If there is not a natural way to color graph nodes, then let $c_i = 0$ for all $i$. We use $W_i^\ell = (j_1 = i, \ldots, j_\ell)$ to denote a walk that starts from $i$ and has length $\ell$. Note that a walk allows repetition of nodes and edges. Let $\Omega_i^\ell$ denote the set of all such walks.

In a graph $S$, we often designate a special node $i \in V(S)$ as the *anchor* of $S$ and then get an anchored graph $S_i$. If $S$ is from a larger graph $G$, we also call $S_i$ an anchored subgraph. We say two anchored subgraphs $S_i$ and $S'_j$ are *isomorphic*, $S_i \cong S'_j$, if there exists a ismorphic mapping from $S$ to $S'$, and the mapping maps $i$ to $j$.

**The 1-WL algorithm.** [1] The 1-WL algorithm plays an important role in the detection of graph isomorphism. Formally, the 1-WL algorithm runs multiple "message-passing" rounds to color graph nodes. Suppose each node $i \in G$ initially has a color $c_i^0 = c_i$. Then in each round $k$ each node get a new color:

$$c_i^k := \left( c_i^{k-1}, \left\{\!\!\left\{ c_j^{k-1} : (i, j) \in E(G) \right\}\!\!\right\} \right) \tag{1}$$

Here $\{\!\!\{\cdot\}\!\!\}$ denote a multiset, a set that allows duplicate elements. The tuple, which contains $i$'s previous color and its neighbors' previous colors, is hashed to a new color $c_i^k$. Note that $c_i^k = c_i^k \Rightarrow$

---

[1]To be exact, this is the color-refinement algorithm, which is slightly different from the 1-WL algorithm (Grohe, 2021). But we follow the literature and still refer it as the 1-WL algorithm.

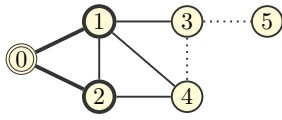

Figure 1: Examples of anchored graphs. $S_0^0$ is a singleton, $S_0^1$ is indicated by very thick lines, $S_0^2$ also includes thick lines, and $S_0^3$ includes the entire graph.

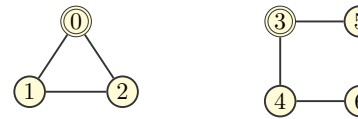

Figure 2: Two structurally different graph nodes (0 and 3) get the same color at every coloring round.

$c_i^{k-1} = c_{i'}^{k-1}$, so the 1-WL algorithm never reduces the number of colors used in each round. The algorithm converges when the number of colors no longer increases. We use $c_i^\infty$ to denote the color of a node $i$ at convergence.

If two graphs $G$ and $G'$ are isomorphic, then they get the same multiset of colors, $\{\{c_i^\infty : i \in V\}\} = \{\{c_{i'}^\infty : i' \in V'\}\}$. But there are also cases in which non-isomorphic graphs $G \not\cong G'$ still have $\{\{c_i^\infty : i \in V\}\} = \{\{c_{i'}^\infty : i' \in V'\}\}$.

The 1-WL algorithm is an appealing method to extract structural information because of its efficiency (Shervashidze et al., 2011). It only runs a small number of rounds to converge and only needs linear time in the number of edges in each round.

**Message-passing neural networks (MPNNs).** An MPNN takes a graph as the input and passes messages between nodes to learn node representations. It has a similar algorithmic structure as 1-WL but use vectors to represent node "colors". Suppose each node $i$ has an initial feature vector $\mathbf{z}_i^0$. Then in each message-passing round,

$$\mathbf{z}_i^k = \text{update}\left(\mathbf{z}_i^{k-1}, \text{aggregate}\left(\{\{\mathbf{z}_j^{k-1} : j \in \mathcal{N}(i)\}\}\right)\right) \tag{2}$$

Here $\text{aggregate}(\cdot)$ aggregates representation vectors in a multiset into a single vector, and the $\text{update}(\cdot)$ computes the new representation of $i$. The two functions need to be differentiable. They together imitate the hashing function in (1).

The expressiveness of an MPNN is bounded by the 1-WL algorithm: if two nodes get the same color $c_i^\ell = c_j^\ell$ after $\ell$ 1-WL rounds, then their representations $\mathbf{z}_i^\ell = \mathbf{z}_j^\ell$ must be the same.

## 4 METHOD

In this work, we consider the structural information at a graph node. Here we consider node colors computed by the 1-WL algorithm, given that it is a clean computation model for a GNN.

### 4.1 ANCHORED SUBGRAPHS

We first precisely define what information we need to encode in a node's representation. To do so, we identify the "receptive" field of a node color $c_i^\ell$ computed at the $\ell$-th 1-WL round. With $\ell$ round of message passing, a node's information can be passed along any walk with length $\ell$. Therefore, the color $c_i^\ell$ of node $i$ at round $\ell$ encodes the information from the anchored subgraph $S_i^\ell$:

$$S_i^\ell = \dot{\cup}_{W_i^\ell \in \Omega_i^\ell} W_i^\ell \tag{3}$$
$$= (\{j : \text{dist}(i, j) \le \ell\}, \{(j, k) \in E : \text{dist}(i, j) < \ell, \text{dist}(i, k) \le \ell\}). \tag{4}$$

Here we treat a walk as a collection of edges, and $\dot{\cup}$ takes unique edges in walks $\Omega_i^\ell$ to form the subgraph $S_i^\ell$.

The anchored subgraph $S_i^\ell$ is the receptive field of $c_i^\ell$: the calculation of $c_i^\ell$ does not involve any nodes or edges beyond $S_i^\ell$. At the same time, each node or edge in $S_i^\ell$ may affect the color $c_i^\ell$. The superscript $\ell$, which we call the *order* of the anchored subgrah, indicates the length of walks composing $S_i^\ell$. The radius of $S_i^\ell$ is at most $\ell$ and can be smaller when the graph $G$ is smaller.

Figure 1 shows one example of $S_i^\ell$ for different $\ell$ values: $S_0^0$ is the singleton with node 0, $S_0^1$ is indicated by very thick lines, $S_0^2$ is expanded to include thick lines, and $S_0^3$ is further expanded to include the entire graph. Note that $S_i^\ell$ does not contain edges between $j$ and $k$ if they are both at distance $k$ from the anchor $i$.

From the definition, we also have

$$S_i^\ell = \dot{\cup}\{(i, W_j^{\ell-1}) : W_j^{\ell-1} \in \Omega_j^{\ell-1}, j \in \mathcal{N}(i)\}$$

$$= \dot{\cup}\left(\cup_{j \in \mathcal{N}(i)}\{(i,j)\} \cup S_j^{\ell-1}\right). \tag{5}$$

Here $(i, W_j^{\ell-1})$ is a length-$\ell$ walk with $i$ as the starting node and the rest specified by $W_j^{\ell-1}$. The equation means that $S_i^\ell$ consists of anchored subgraphs $\{S_j^{\ell-1} : j \in \mathcal{N}(i)\}$ and edges connecting $i$ to the anchors of these subgraphs.

One fundamental questions is how much information about $S_i^\ell$ is encoded in $c_i^\ell$. The previous analysis of the 1-WL algorithm shows that

$$S_i^\ell \cong S_j^\ell \Rightarrow c_i^\ell = c_j^\ell. \tag{6}$$

In the opposite direction, the color $c_i^\ell$ cannot uniquely decide the anchored subgraph $S_i^\ell$, and Figure 2 shows one example. However, $c_i^\ell$ does encode rich information about the structure surrounding $i$. The neural representation $\mathbf{z}_i^\ell$ learned by a GNN also has receptive field $S_i^\ell$ and contains information about it. In a learning task, if a node's label is mostly decided by the graph structure surrounding the node, then $\mathbf{z}_i^\ell$ can be used to learn such relationships and predict node labels.

### 4.2 BFS-Trees and WL-trees

It is clear that a node color $c_i^\ell$ cannot encode all information of the anchored subgraph $S_i^\ell$. In this section, we propose to examine the information contained in $c_i^\ell$ with a tree form of 1-WL colors. In particular, we construct a tree from a BFS on $G$ starting from one of its nodes. To make the discussion easier, we refer $G$'s nodes by their ids, which are their indices in $V(G)$.

We construct a *BFS-tree* from a BFS starting from $i \in G$ and record node ids to not lose information. Here we allow the BFS to revisit previous nodes *except* the direct parent.

**Definition 1.** *A BFS-Tree $B_i^\ell$ at a node $i \in G$ is recursively constructed as follows:*

    *i) $B_i^0$ is a singleton with the root labeled with id $i$ and color $c_i$;*

    *iii) for each tree node $t$ at level $\ell' < \ell$, let $t$'s id be $j$. Let $i$ be $t$'s parent's id if $t$ has a parent or -1 otherwise. For each neighbor $j' \in \mathcal{N}(j)\backslash i$, create a child with id $j'$ and color $c_{j'}$.*

Let $V(B_i^\ell)$ denote the node set of the BFS-tree. We use a function $\mathrm{id}(\cdot) : V(B_i^\ell) \to V(S_i^\ell)$ to map a tree node to a node in $S_i^\ell$. Note that a node in $S_i^\ell$ may appear multiple times in the tree. Figure 3 (middle) shows one example of BFS-tree derived from the anchored subgraph on the left. Our BFS-tree is different from the roll-out tree considered by Shervashidze et al. (2011); Zhang & Li (2021); Jegelka (2022) – here a tree node does not include its parent as its child.

A BFS-tree $B_i^\ell$ preserves all information about $S_i^\ell$. Since $B_i^\ell$ is obtained by a BFS with $\ell$ levels, all edges in walks in $\Omega_i^\ell$ are in $B_i^\ell$. Then we can recover $S_i^\ell$ by merging $B_i^\ell$'s tree nodes by their ids.

From a BFS-tree $B_i^\ell$, we can read out other shallower BFS-trees.

**Lemma 2.** *Let $t \in B_i^\ell$ be a tree node at level $d$, and $j = \mathrm{id}(t)$. By changing the root of the BFS-tree to be $t$ and pruning nodes below level $\ell - d$, the resultant tree is the BFS-tree $B_j^{\ell-d}$. As a special case $j \in \mathcal{N}(i)$, the resultant tree is $B_j^{\ell-1}$.*

Now we define a *WL-tree* by dropping node ids from a BFS-tree. Figure 3 (right) shows a WL-tree derived from the BFS-tree in the middle.

**Definition 3.** *A WL-tree $T_i^\ell$ is a rooted tree obtained by dropping node ids of a BFS-tree $B_i^\ell$.*

Here the subscript $i$ indicates that it is computed at node $i$ in the original anchored subgraph $S_i^\ell$. From a WL-tree, we can similarly read out shallower WL-trees possibly at a different node. Figure 4 shows two examples, from which we will show the equivalence of a WL-tree and a 1-WL color.

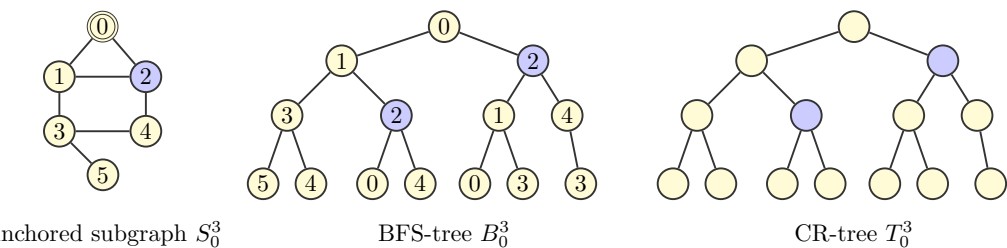

Figure 3: An anchored subgraph, its BFS-tree, and its WL-tree. The anchor node is indicated by double circle. A node's color is either yellow or blue.

**Lemma 4.** *For any WL-tree $T_i^\ell$ obtained from an anchored graph $S_i$, let $t \in T_i^\ell$ be a tree node at level $d$, and $t$ is obtained from a node $j \in S_i$. By changing the root of the BFS-tree to be $t$ and pruning nodes below level $\ell - d$, the resultant tree is the WL-tree $T_j^{\ell-d}$. As a special case $j \in \mathcal{N}(i)$, the resultant tree is $T_j^{\ell-1}$.*

We are ready to show that WL-trees are equivalent to colors from the 1-WL algorithm.

**Theorem 5.** *Let $c_i^\ell$ and $T_i^\ell$ be the 1-WL color and the WL-tree computed from a arbitrary anchored subgraph $S_i^\ell$, then the mapping $c_i^k \leftrightarrow W_i^k$ forms a bijection.*

Here we only show the main idea of the theorem and put the formal proof in the appendix. A node's color $c_i^\ell$ is obtained from its color and its neighbors' colors in the previous round. These colors can be read out from subtrees in the WL-tree $T_i^\ell$ from Lemma 4. Figure 4 shows subtrees $T_0^{\ell-1}$, $T_1^{\ell-1}$, and $T_2^{\ell-1}$, which are equivalent to $c_0^{\ell-1}$, $c_1^{\ell-1}$, and $c_2^{\ell-1}$ respectively. From their equivalency, we can establish the equivalency between $T_0^\ell$ and $c_0^\ell$.

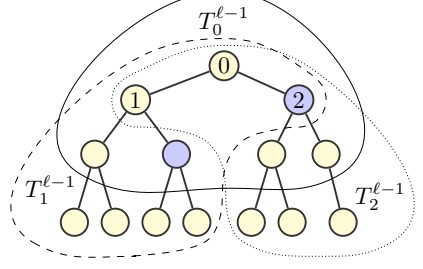

Figure 4: Colors in the $\ell - 1$ round are read out from WL-trees of the order $\ell - 1$.

Similar to the 1-WL algorithm, WL-trees can also be used to distinguish graph structures.

$$T_i^\ell \neq T_j^\ell \Rightarrow S_i^\ell \neq S_j^\ell. \qquad (7)$$

Although there is an bijective mapping between 1-WL colors and WL-trees, WL-trees make the underlying structures easier to analyze. In particular, WL-trees provides a more fine-level analysis of the expressiveness of node representations learned by GNNs.

If an anchored graph is a tree rooted at the anchor, then the WL-trees at the root have the same structure as the anchored graph expect a possible depth difference. It also indicates that any rooted tree can be a WL-tree.

**Theorem 6.** *If an anchored graph $S_i$ is a tree rooted at $i$ and has depth $\ell$, then*

    *i) the WL-tree $T_i^{\ell'}$ with $\ell' < \ell$ is the tree obtained by pruning nodes at levels deeper than $\ell'$ ;*

    *ii) the WL-tree $T_i^\ell$ is $S_i$ with its root at $i$;*

    *iii) the WL-tree $T_i^{\ell'}$ with $\ell' > \ell$ is also $S_i$ with its root at $i$.*

If a graph contains at least one cycle, then its WL-tree can have any depth.

**Corollary 7.** *If an anchored graph $S_i$ has a cycle, then for any $\ell$, at least one leaf node in the WL-tree $T_i^\ell$ is at depth $\ell$.*

If we run enough 1-WL rounds, we can always distinguish a loopy graph from a tree.

**Corollary 8.** *Let $S_i$ be graph with a cycle, and let $S_{i'}'$ be a tree with depth $k$. Then the $k + 1$-th 1-WL round is able to distinguish $i$ and $i'$ from their anchored graphs, that is, the two WL-trees $T_i^{k+1} \not\cong T_{i'}^{k+1}$ obtained at $i$ and $i'$ are different.*

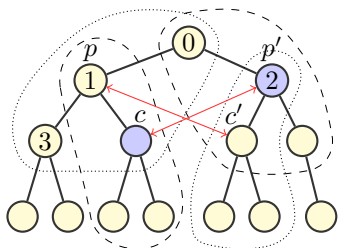 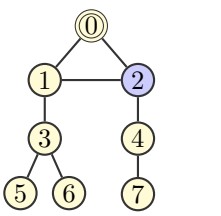 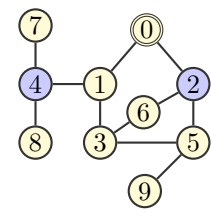

Figure 5: The situation of assigning an existing id 2 to the tree node $c$.

Figure 6: Two anchored graphs graphs recovered the WL-tree.

### 4.3 RECOVERING ANCHORED GRAPHS FROM WL-TREES

A important approach of understanding the information encoded in a WL-tree (or a 1-WL color) is to identify anchored subgraphs compatible with a given WL-tree. In this section, we assume that the given WL-tree is $T_0^\ell$ – the subscript indicates that the root is to be mapped to a node 0 later.

We first consider the simple case that the underlying anchored graph $S_0$ is a tree.

**Corollary 9.** *Let $T_0^\ell$ be a WL-tree with depth $d \leq \ell$, and $T_0^\ell$ is from an anchored graph $S_0$, then:*

    *i) if $S_0$ is known to be a tree with depth $d' \leq \ell$, then $S_0$ has the same structure as $T_0^\ell$;*

    *ii) if $d < \ell$, then $S_0$ must be a tree and have the same structure as $T_0^\ell$.*

Now we consider a general algorithm to identify an anchored graph $S_0$ compatible with a WL-tree $T_0^\ell$. We follow the BFS order and label each tree node with an id, then we get a BFS-tree and then can find an $S_0$.

We first assign 0 as the root's id, then our next steps face the same problem: how to label a child of a parent that has received an id. Suppose $(p, c)$ is a parent-child pair in the tree, and $p$ has id $\mathrm{id}(p)$. Then we need to decide $\mathrm{id}(c)$, which will introduce an edge $(\mathrm{id}(p), \mathrm{id}(c))$ to the anchored graph $S_0$.

There are two cases of assigning $\mathrm{id}(c)$, which can be either a new id or an id used by a previous node $p'$. In the first case, we add a new node $\mathrm{id}(c)$ to the graph and connect it to $\mathrm{id}(p)$. Since $c$ is the first appearance of the graph node $\mathrm{id}(c)$, we need to further label $c$'s descendants in the tree.

The second case is to assign $\mathrm{id}(c)$ with an existing id $\mathrm{id}(p')$, which is used by a previously labeled tree node $p'$. This case is more restricted by the tree structure. Labeling $\mathrm{id}(c) = \mathrm{id}(p')$ means that $c$ and $p'$ correspond to the same graph node $\mathrm{id}(p')$ in the anchored graph. Figure 5 shows one example of this situation. If we need to assign 2 as the id of $c$, then $c$ and $p'$ are from the same graph node 2, and then they need to have matching WL-trees (indicated by dashed circles). Correspondingly, $p'$ needs to have a neighbor $c'$ that matches $p$. WL-trees (indicated by dotted circles) of $c'$ and $p$ need to match as well.

Formmally, the assignment $\mathrm{id}(c) = \mathrm{id}(p')$ needs satisfy the following condition.

**Condition 1.** *Let $(p, c)$ be a parent-child pair in a WL-tree $T_0^\ell$. Suppose $p$ has id $\mathrm{id}(p)$, and another node $p'$ in the tree has id $\mathrm{id}(p')$. Let $1 \leq k \leq \ell$ be the level of $c$, and $k' \leq k$ be the level of $p'$. In order to label $\mathrm{id}(c) = \mathrm{id}(p')$, the following must hold:*

    *i) $p'$ is not the parent or a sibling of $c$, and $p'$ has a neighbor $c'$ with no id yet;*

    *ii) the WL-tree rooted at $c$ is isomorphic to the WL-tree rooted at $p'$ with depth $\ell - k$;*

    *iii) the WL-tree rooted at $c'$ is isomorphic to the WL-tree rooted at $p$ with depth $\ell - k' - 1$.*

If this condition is satisfied, it means that the WL-tree at graph node $\mathrm{id}(c)$ is the same as the WL-tree at $\mathrm{id}(p')$ up to the known depth $\ell - k$, so $c$ and $p'$ can be from the same graph node. Similarly, $c'$ and $p$ can be from the same graph node. Then we can set $\mathrm{id}(c) = \mathrm{id}(p')$ and $\mathrm{id}(c') = \mathrm{id}(p)$, and at the same time add an edge $(\mathrm{id}(p), \mathrm{id}(p'))$ to the subgraph $S$. The descendants of $c$ or $c'$ will not be further labeled because their ids are decided by subtrees rooted at $p$ and $p'$.

---

**Algorithm 1** Compute an anchored subgraph from a WL-tree

---

**Require:** A WL-tree `T`
 1: Initialize a graph `S` as a singleton with id `gid = 0`
 2: Create a queue `Q` and set `Q = children(root(T))`
 3: **while** `Q` is not empty **do**
 4:     `c = Q.pop()`
 5:     **if** `use_existing_id` **and** can find `(p2, c2)` satisfying Condition 1 **then**
 6:         Set `id(c) = id(p2)`, `id(c2) = id(parent(c))`
 7:         Add an edge `(id(parent(c)), id(c))` to `S`
 8:         `Q.remove(c2)`
 9:     **else**
10:         `gid += 1`, and set `id(c) = gid`
11:         Add a node `id(c)` and an edge `(id(parent(c)), id(c))` to `S`
12:         `Q.push(children(c))`
13:     **end if**
14: **end while**
15: **return** `S`

---

A BFS labeling order guarantees that the node with a new id has a deeper subtree than later nodes taking the same id. The algorithm is summarized in Algorithm 1. It can optionally decide `use_existing_id` in each "while" loop to get an anchored graph $S_0$.

Figure 6 shows two other anchored graphs recovered from the WL-tree. Besides the graph in Figure 3 (left) and these two graphs, there are 5 more trees corresponding to the WL-tree in Figure 5.

**Theorem 10.** *With $T_0^\ell$ as the input, Algorithm 1 returns an anchored graph $S_0$ that has WL-tree $T_0^\ell$.*

We can further extend Algorithm 1 to enumerate all anchored graphs for one WL-tree. In particular, we need to enumerate combinations of choices of $\mathrm{id}(c)$ at different $c$. We also need to identify isomorphic anchored graphs and only keep one of them. Here we omit the implementation details. The computation is much higher than identifying one anchored graph, but we want to emphasize is that this algorithm is only used to diagnose the 1-WL algorithm or a GNN, and the computation is not a main consideration here.

## 5 INFORMING SUBGRAPH MATCHING

A WL-tree not only carries information about an anchored graph $S_i$ but also indicates whether $S_i$ contains a smaller anchored subgraph $\hat{S}_i \subseteq S_i$.

**Theorem 11.** *Suppose $\hat{S}_i$ is a subgraph of $S_i$ and also has anchor $i$. Let $T_i^\ell$ and $\hat{T}_i^\ell$ be the respective $\ell$-th order WL-trees of $S_i^\ell$ and $\hat{S}_i$, then a subtree of $T_i^\ell$ sharing the same root matches $\hat{T}_i^\ell$.*

We derive this fact from the BFS-tree $B_i^\ell$ of $S_i^\ell$: we only need to take the subtree with nodes in the set $V(\hat{S}_i)$, then the resultant subtree is the BFS-tree of $\hat{S}_i$, and then the subtree without ids is the WL-tree of $\hat{S}_i$.

This theorem indicates that $\hat{T}_i^\ell \nsubseteq T_i^\ell \Rightarrow \hat{S}_i \nsubseteq S_i^\ell$, which can be used to execlue some matching possibilities. This type of information may have already been potentially used in graph learning models for subgraph matching (Ying et al., 2020; Bai et al., 2021; Li et al., 2019). While neural representations hardly give any guarantee over a possible matching, a WL-tree can firmly exclude some matching possibilities. An interesting topic is to use WL-trees to direct matching decisions.

## 6 AN ANALYSIS OF COLORING STRATEGIES THAT IMPROVE MPNN'S EXPRESSIVENESS

In this section, we analyze two GNNs, CLIP (Dasoulas et al., 2021) and Nested GNN (Zhang & Li, 2021), which feed extra node inputs to improve the expressiveness. A CLIP uses random colors to

distinguish nodes with the same attributes and then run a GNN to compute node representations. By pooling representations learned from permutations of colors, the learned representation eliminates the variance brought by different color codings. Similar ideas of using random inputs to improve the expressiveness of GNNs are also explored by GNN-RNI (Abboud et al., 2020) and rGNN (Sato et al., 2021). CLIP has one issue that it requires too many colors when the graph is large. Here we consider a variant of CLIP, termed CLIP-2, that adds random binary colors to node inputs and computes node representations.

Nested GNN run two layers of GNN. The inner-layer GNN runs on a node's neighborhood to compute the node's feature input. Note that a node's neighborhood is the subgraph induced by the node's neighbors within $h$ hop. The computed inputs are able to distinguish some graph nodes than cannot be distinguished by the 1-WL algorithm. Then the outer-layer GNN takes these new node inputs and compute node representations with normal-message passing layers. The learned node representations are able to encode more information than those from a vanilla GNNs.

Node representations are usually real numbers, which pose difficulties to analysis of the structural information encoded. Our plan here is to find algorithmic models for these GNN variants, just like the 1-WL algorithm for a vanilla GNN, and then check a node's representation in discrete form.

CLIP-2 directly assign random colors to graph nodes. In a NGNN with center pooling, the representation computed by the inner-layer GNN can also be viewed as colors: they are equivalent to colors obtained by running the 1-WL algorithm on the neighborhood of the node. Then we can compare the two GNNs against WL-trees.

**Theorem 12.** *Given a graph $G$, suppose a CLIP-2 uses a set $\mathcal{C}$ of colorings of $G$ and an $\ell$-layer GIN to compute a node $i$'s representation:*

$$\bar{\mathbf{h}}_i = \max_{\bar{c} \in \mathcal{C}} \mathrm{GIN}(i, G, \bar{c}), i \in V(G) \tag{8}$$

*The WL-tree $\bar{T}_i^\ell$ at $i$ is computed from $(G, \bar{c})$ with a coloring $\bar{c} \in \mathcal{C}$. Then $\bar{T}_i^\ell = \bar{T}_{i'}^\ell \Rightarrow \bar{\mathbf{h}}_i = \bar{\mathbf{h}}_{i'}, i, i' \in V(G)$.*

**Theorem 13.** *Given a graph $G$, suppose a NGNN first uses an $\ell_1$ layer GIN to compute each node $i$'s representation from its $h$-hop neighborhood $G(i, h)$, and then uses another $\ell$-layer GIN to compute the node representation:*

$$\hat{\mathbf{x}}_j = \mathrm{GIN}_I(j, G(j, h), c), j \in V(G) \tag{9}$$

$$\hat{\mathbf{h}}_i = \mathrm{GIN}_O(i, G, \{\hat{\mathbf{x}}_j, j \in V(G)\}), i \in V(G). \tag{10}$$

*Corresponding to $\mathrm{GIN}_I$, let $T_j^{\ell_1}$ be the WL-tree computed from $(G(j, h), c)$. Let $\hat{T}_i^\ell$ be the WL-tree at $i$ and computed using WL-trees as graph colors, $\hat{c}_j = T_j^{\ell_1}$. Then $\hat{T}_i^\ell \leftrightarrow \hat{\mathbf{h}}_i$ is a bijection.*

With these two theorems identifying the two GNNs' computation models, now we use WL-trees to assess their expressive power. In particular, we check their abilities in identifying the underlying subgraphs. We first count the number of subgraphs that can be possibly identified from a WL-tree. We also take data statistics into account and calculate the conditional entropy $\mathbb{H}[S_i^\ell | T_i^\ell]$ from the data. A smaller count or conditional entropy means that the WL-tree can better identify a node's surround structure.

We conduct the evaluation on three datasets with low node degrees so we can afford counting possible anchored subgraphs. MUTAG (Debnath et al., 1991) is a collection of 188 chemical compounds, which are represented as graphs. On average each graph has 17.9 nodes and 19.8 edges. Each node takes one from 7 possible colors. Road-MN (Rossi & Ahmed, 2015) contains the road network of Minnesota, which contains 2.6K nodes and 3.3K edges. We set color 0 to all nodes. The CiteSeer dataset (Giles et al., 1998) consists of 3312 scientific publications and 4732 citation links. We treat it as an undirected graph and remove all node attributes. We further set all node with color 0.

For each node $i$ in the graph, we extract $S_i^\ell$, $T_i^\ell$, $\bar{T}_i^\ell$, and $\hat{T}_i^\ell$. Here $S_i^\ell$ and $T_i^\ell$ use the original graph's node colors. $\bar{T}_i^\ell$ uses node colors that combine the original node color and random colors. For $\hat{T}_i^\ell$, node colors are hashing of the inner layer WL-trees, as described in Theorem 13. We choose $h = 1$ and $\ell_1 = 2$ in the calculation of these WL-trees.

We first check the number of possible anchored subgraphs that are compatible with three types of WL-trees. In this experiment, we set the depth of WL-trees to 3. For each node's WL-tree, we

|          | MPNN          | CLIP-2        | NGNN         |
|----------|---------------|---------------|--------------|
| MUTAG    | $11.5 \pm 26.0$ | $3.1 \pm 5.1$  | $1.9 \pm 1.8$ |
| Road-MN  | $562 \pm 3410$ | $148 \pm 1276$ | $17 \pm 213$ |
| CiteSeer | $267 \pm 1402$ | $235 \pm 1609$ | $230 \pm 1751$ |

Table 1: Average counts of anchored subgraphs corresponding to each WL-tree.

| depth  | 3     | 4     | 5     | 6    |
|--------|-------|-------|-------|------|
| MPNN   | 0.15  | 0.078 | 0.061 | 0.02 |
| CLIP-2 | 0.004 | 4e-4  | 4e-4  | 0    |
| NGNN   | 0.035 | 0.013 | 0.012 | 6e-4 |

Table 2: The conditional entropy of anchored subgraphs on the MUTAG dataset.

| depth  | 3     | 4     | 5     | 6     |
|--------|-------|-------|-------|-------|
| MPNN   | 0.39  | 0.09  | 0.02  | 0.003 |
| CLIP-2 | 5e-4  | 0     | 0     | 0     |
| NGNN   | 0.008 | 0.003 | 5e-4  | 1e-4  |

Table 3: The conditional entropy of anchored subgraphs on the Road-MN dataset

| depth  | 3     | 4     | 5      | 6      |
|--------|-------|-------|--------|--------|
| MPNN   | 0.056 | 0.011 | 0.0061 | 0.0061 |
| CLIP-2 | 8e-4  | 4e-4  | 0      | 0      |
| NGNN   | 5e-4  | 0     | 0      | 0      |

Table 4: The conditional entropy of anchored subgraphs on the CiteSeer dataset.

count the number of possible anchored subgraphs behind it. The results are shown in Table 1. From the results, we see that extra node colors from CLIP-2 and NGNN significantly reduces the number of graphs corresponding to each node's WL-tree. These counts are highly unbalanced at different nodes and thus introduce high variance. A WL-tree from a densely connected neighborhood tend to have many more anchored subgraphs than those from sparse neighborhoods.

We then check the conditional entropy $\mathbb{H}[S_i^\ell | T_i^\ell]$, which includes data statistics in consideration. For each dataset, we vary $\ell$ from 3 to 6 and check conditional entropies obtained from three coloring methods. The results from the three datasets are in Table 2, 3, and 4. We see that CLIP-2 and NGNN both reduce the conditional entropy when $\ell = 3$. However, when $\ell$ is large, all conditional entropy values are small. If the anchored subgraphs corresponding to a WL-tree are equally possible, then conditional entropy would be $\log(C)$ with $C$ being the count from Table 1, and the conditional entropy would be much larger. However, in the data the probability concentrates to a few anchored subgraphs corresponding to a WL-tree. Also given the large number of subgraph patterns, the reported numbers underestimate the true entropy in an inductive setting (e.g. MUTAG), but it is less a problem in transductive setting (with only one network). When the conditional entropy is small, a GNN has the ability to differentiate nodes by their WL-trees at least on the training set. This is true particularly with a large depth $\ell$.

**Discussion.** Now we'd like to re-consider the node classification task after the above investigation. We categorize the error source in three layers. First, nodes with different labels may have the same anchored graphs. For example, two communities have the same structure but different hobby labels. In this situation, the model needs to consider spatial similarity besides structural similarity, e.g. by including spectral information (Li et al., 2020; Dwivedi et al., 2021; Lim et al., 2022). Second, nodes with different labels have the same WL-tree but different anchored subgraphs. This kind of error is indeed due to a GNN's inability in distinguishing nodes by their WL-trees. In terms of reducing the this type of error, there seem to be a big gap between graph theory and practice. If we consider the worst case, then a GNN doesn't seem to be able to fully represent all anchored subgraphs given its linear running time. But if we also consider data statistics, extra colors by CLIP and NGNN have provided satisfying differentiation of nodes with different surrounding structures. Third, nodes with different labels can be discriminated by WL-trees but not by a GNN. The over-smoothing (Rusch et al., 2023) and over-squashing (Topping et al., 2021) issues are all at this level. Our new analytic tool suggests that the solution to this issue boils down to better representation learning of WL-trees. The prospective learning architecture might be very different from message-passing used by GNNs.

## 7 CONCLUSION

In this work, we have developed WL-trees as a new tool for the analysis of node representations learned by GNNs. Compared with previous computation models of GNNs, WL-trees provides a deeper understanding of graph structures that can be encoded by node representations. Our new analysis also points to new directions of learning these representations.

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
