# A    APPENDIX

Proof of Lemma 2.

*Proof.* Suppose the root of the tree is $\tau$, and $\mathrm{id}(\tau) = i$. Let $t$ be a tree node at level 1, and $j = \mathrm{id}(t)$. Then $j \in \mathcal{N}(i)$ in $G$. We consider a rotation operation and move $t$ up to be the root, then $\tau$ a child of $t$. Now all $t$'s children correspond to $\mathcal{N}(j)$. After rotation, $t$ becomes the parent of $\tau$. Now $\tau$'s children correspond $i$'s neighbors except $j$. All other tree nodes do not need rearrangement because their parent do not change.

In the new tree, the bottom two levels of decendents of $\tau$ are now at level $\ell$ and $\ell + 1$. Then we prune these decendents and get a tree with depth $\ell - 1$. This new tree satisfies the definition of the BFS-tree, and thus is the BFS-tree with depth $(\ell - 1)$ obtained at $j \in G$.

For a node $t$ at level $k$, then we need to rotate its $k - 1$ ancestors to be roots, and then rotate $t$ to be the root. Each rotation reduces the depth of the tree by 1. After $k$ rotations, we will get the BFS-trees $B_j^{\ell - k}$.

$\square$

Proof of Lemma 4

*Proof.* Suppose we move a tree node $t$ to be the root of $T_i^\ell$. The WL-tree $T_i^\ell$ must be derived from a BFS-tree. Suppose $t$ corresponds to a node $t'$ in the BFS-tree. Then moving $t'$ gives a new BFS-tree, which has the same structure the tree obtained by moving $t$ to be the root, so the latter one is the WL-tree derived at $\mathrm{id}(t')$. $\square$

Proof of Theorem 5.

*Proof.* We construct the bijection by induction. We consider $\ell = 0$ in the base case: an anchored graph $S_i^0$ is a singleton with the node in color $c_i$, then $i$ gets color $c_i^0 = c_i$ from the 1-WL algorithm, and the CR-tree $T_i^0$ is also a singleton in color $c_i$. The bijection maps a color $c_i^0$ to a singleton in the same color.

Then we assume that the statement is true for $\ell$, then we show that such an injective mapping exists for $\ell + 1$. That is, if both $T_i^{\ell+1}$ and $c_i^{\ell+1} = (c_i^\ell, \{\{c_j^\ell : (i,j) \in E\}\})$ are both computed from an anchored subgraph $S_i^{\ell+1}$, then we can read out $T_i^{\ell+1}$ or $c_i^{\ell+1}$ from the other. We first consider mapping $T_i^{\ell+1}$ to the color $c_i^{\ell+1}$. By the assumption, the color $c_i^\ell$ can be read from $T_i^\ell$ from the assumption. For each $j \in \mathcal{N}(i)$, the color $c_j^\ell$ is computed from $S_j^\ell$. From the Lemma above, we can identify all $T_j^\ell$-s for all $S_j^\ell, j \in \mathcal{N}i$. Then we map each $W_j^k$ to $c_j^k$ for each $j \in \mathcal{N}i$ by the assumption. With all these components we can map $W_i^{k+1}$ to $c_i^{k+1}$.

Now we consider the mapping from $c_i^{k+1}$ to $W_i^{k+1}$. From $c_i^k$, we have $W_i^k$, which is the CR-tree with depth $k$. We only need to expand one more level from $W_i^k$. For each leaf node at level $k$ in $W_i^k$, it must appear in one of $\{W_j^k\}$ at level $k-1$. We can find all its neighbor colors there $\{\{c_0, \ldots, c_k\}\}$. Suppose the color of parent is $c_p$, then we just creat nodes as children of this node. These children take colors $\{\{c_0, \ldots, c_k\}\} \setminus c_p$. $\square$

Proof of Theorem 6

*Proof.* In the construction of the BFS-tree, a node can only have children corresponding to children from $S_i$. Therefore, the BFS-tree with depth $\ell' = \ell$ is the same as $S_i$. If $\ell' < \ell$, then the BFS-tree can only include the top $\ell'$ levels of $S_i$. When $\ell' > \ell$, a leaf node cannot expand to any children in the BFS-tree, the BFS-tree will still be $S_i$. $\square$

Proof of Corollary 7

*Proof.* For each node in a cycle in the graph $S_i$, the corresponding tree node can always add at least one node as its child. The child can be expanded in the same way, so the depth of the WL-tree can be arbitrarily deep. □

Proof of Corollary 8

*Proof.* With Theorem 6 and Corollary 7, $T_i^{k+1}$ and $T_{i'}^{k+1}$ cannot have the same depth, so they cannot be isomorphic. □

Proof of Corollary 9

*Proof.* We only need to reverse the argument in Theorem 6 to show part i). If $S_0$ is a tree, then it has the same structure as its WL-tree, so $S_0$ has the same structure as $T_0^\ell$. For ii) we only to use the conclusion from Corollary 7. If $\ell > d$, then $S_0$ must be a tree, and then by i), $S_0$ must be $T_0^\ell$.

□

Proof of Theorem 10

*Proof.* The algorithm must converges because a tree node appears at most once in the queue.

After the labeling procedure is done. Suppose $\mathrm{id}(c) = \mathrm{id}(p')$, we can match the subtree rooted at $c$ to a subtree rooted at $p'$. $p'$ must be the first appearance of $\mathrm{id}(p')$, so its children must be labeled. We can copy $(p')$'s children's ids to $c$'s children according to the matching. By applying this operation recursively, all nodes will be labeled. When two nodes take the same id, they will have same neighbor set. So this tree is consistent with the definition of a BFS-tree, so we can recover an anchored graph from it.

□

Proof of Theorem 12

*Proof.* Given the same input, the GIN computes representations equivalent with 1-WL colors (Xu et al., 2018) as well as WL-trees. Since the set of colorings is known, we can permute node colors of WL trees to get all WL-trees that can be compute from all colorings in $\mathcal{C}$. Therefore, We can identify GIN outputs from a WL-tree.

□

Proof of Theorem 13

*Proof.* The GIN computes representations that are equivalent to 1-WL colors (Xu et al., 2018), and thus they are also equivalent to WL-trees. So there is an injective mapping between the inputs to the outer layer GNN and colors used by the 1-WL algorithm.

At the same time, the outer-layer GNN also has the same ability as the 1-WL algorithm, so the injective mapping is able to be maintained across GNN layers/1-WL iterations. Therefore, the final output will be equivalent to 1-WL colors and WL-trees.

□