# OpenReview forum: "WL-Tree: a New Tool for Analyzing Graph Neural Networks"
_ICLR.cc/2024/Conference — Submitted to ICLR 2024_

### Official Review · Reviewer_py5L · 2023-10-13

**Soundness:** 2 fair
**Presentation:** 3 good
**Contribution:** 1 poor
**Rating:** 3
**Confidence:** 5

**Summary:**

The paper presents WL-tree, a tool for analyzing graphs based on the multiset of walks of a given length that leave a node. An algorithm that identifies whether a certain node satisfies a given WL tree is also presented. Finally, a working implementation of two graph neural networks (GNNs) that enhance the expressiveness of message-passing GNNs with node id representations.

**Strengths:**

The presentation is good.

**Weaknesses:**

The tool presented in the paper, WL tree, essentially corresponds to the well-notion of tree unravelling from a node in a graph. That this notion is equivalent with WL coloring is absolutely folklore and has been used in many papers for decades. As such, the paper does not bring any new conceptual contribution into the picture. Theoretically speaking, all results in the paper are simple exercises.

The authors also show a poor understanding of the related literature. A concrete example is when they mention that WL has the same expressive power than *guarded* FO_2^\cnt. This is simply not true, and it is not what Cai et al have proved. They have shown that the distinguishing expressive power of WL is exactly the same as FO_2^\cnt (the guarded version is, in fact, weaker). The results by Barceló et al do not concern this notion of expressive power, but a different one. They show that each guarded FO_2^\cnt unary formula can be turned into an equivalent GNN over the set of all graphs. That is, the result by Barceló et al is *uniform*, while the one by Cai et al. is not (and neither is the result of Morris et al.)

**Questions:**

I have no concrete questions. The paper is below the bar in my view.

---

> ### Author Response · Authors · 2023-11-11
> **A quick reponse**
>
> If you state "That this notion is equivalent with WL coloring is absolutely folklore and has been used in many papers for decades." Could you just point out one paper to support your statement? Thank you!
>
> We will post a longer response later.

---

> > ### Comment · Reviewer_py5L · 2023-11-13
> >
> > Sure, the notion of unravelling in relationship with WL has been used, for instance, in the following two papers:
> >
> > - https://arxiv.org/pdf/2204.04661.pdf (See Theorem C.2)
> > - https://openreview.net/pdf?id=r1lZ7AEKvB (see Definition C.2)

---

> > > ### Author Response · Authors · 2023-11-23
> > > **Respectfully Disagree**
> > >
> > > 1. "That this notion is equivalent with WL coloring is absolutely folklore and has been used in many papers for decades"
> > >
> > >
> > > First, our work is not only about the equivalence between WL coloring and WL-trees. We have plenty of new results following the equivalence. Second, you say that the result has been known "for decades", but you only provided publications in the last three years. We hope your statement is well supported by evidence.
> > >
> > > 2.  "the paper does not bring any new conceptual contribution into the picture. Theoretically speaking, all results in the paper are simple exercises."
> > >
> > > We would like to politely point out results that are not in the literature. First, we have shown that a GNN may confuse graph node with a node in a tree. Second, we have devised an algorithm to identify all graph structures behind a WL color. We agree that the first result is an *easy* conclusion from the equivalence, but it is *new* (please provide a citation if you don't think so). For the second contribution, we don't think it is easy, and we also don't know any existing work on this problem. (again please provide citations if you don't agree).

---

### Official Review · Reviewer_UyCd · 2023-10-25

**Soundness:** 1 poor
**Presentation:** 2 fair
**Contribution:** 2 fair
**Rating:** 3
**Confidence:** 4

**Summary:**

In the paper, the authors propose a new tree model for GNNs, which they call WL-trees. The key idea of constructing WL-trees is based on a variant of breadth first search that allows to revisit non-parent nodes. Then the authors show that WL trees are equivalent to the 1-WL algorithm in terms of node coloring. They also propose an algorithm to identify subgraphs anchored at nodes which have the same node representations corresponding to a given WL tree. The contributions claimed by the authors are that the proposed WL trees can provide a more intuitive understanding of graph structures learned by message-passing GNNs.

**Strengths:**

[1] The paper proposes a different perspective to analyse graph structures underlying message-passing graph neural networks.

[2] The connections between their proposed WL trees and anchored graphs are discussed, along with an algorithm that can identify anchored subgraphs corresponding to a given WL tree.

[3] Two existing GNN models are considered and analyzed in the experiments.

**Weaknesses:**

[W1] The proposed method is not well defined. Below are some specific comments:

 - Page 3: The formulations in Equations 3-5 are not consistent. In Equation 3, an anchored subgraph is defined in terms of a set of walks but Equation (5) defines an anchored subgraph as a set of pairs of nodes and walks. Further, the definition of $\dot{\cup}$ is not clearly presented. Also, why $dist(i,j)$ is only less than $\ell$ but $dist(i,k)$ is less than or equal to $\ell$?

- Page 4: For the function id(·) that maps a tree node to a node in an anchored graph, since a node in an anchored graph may appear multiple times in the tree, is it still a function?

[W2] The proposed WL-trees differ from the computational tree structures of message-passing GNNs mainly in disallowing the revisit of the parent nodes. The authors claim that the proposed WL-trees are equivalent to the 1-WL algorithm in terms of node coloring. This does not seem correct. Consider a counter-example, where G is a graph consisting of two triangles and H is a cycle of length 6. These two graphs cannot be distinguished by 1-WL, but would have different WL-trees proposed in the paper.

[W3] The tree structures underlying message-passing GNNs and their connection to 1-WL have been well studied in the literature. It is unclear why the proposed WL-trees can provide a more fine-level analysis of the expressiveness of node representations learned by message-passing GNNs. In particular, the proposed WL-trees are not equivalent to 1-WL (see the above point [2]).

[W4] In what kinds of scenarios will the proposed algorithm 1 be useful?

[W5] For the section 6, what are the justifications for selecting CLIP and Nested GNN? There are a large number of GNN models developed in the literature. I don't see why these two particular GNN models are selected for analysis.

[W6] Theorem 12 and Theorem 13 look confusing. Why is max used in Equation 8? Is the notation $G(j,h)$ defined? Also, the expressive power of Nested GNN goes beyond 1-WL, but the WL-trees proposed in the paper are claimed to be equivalent to 1-WL. So why does Theorem 13 state that there is a bijective mapping between WL-trees and their node embeddings calculated by Equation 10?

[W7] For the statement "A smaller count or conditional entropy means that the WL-tree can better identify a node’s surround structure", is any theoretical justification? Analysing GNN models using average counts of anchored subgraphs and the conditional entropy of anchored graphs look ad hoc.

**Questions:**

W1 - W7

---

> ### Author Response · Authors · 2023-11-23
> **Thank you for your comments**
>
> Thank you for your comments. We appreciate that you have seen our effort in getting a new perspective of graph structures behind GNNs.
>
> --- [W1] Equations 3 and 5.
>
> Equation 3 is the definition, and equation 5 is derived from equation 3. The idea of equation 5 is as follows: if i is the anchor of an anchored subgraph, then the anchored subgraph can be viewed as the union of 1) i and i's neighbors and 2) anchored subgraphs surrounding i's neighbors.
>
> In the edge set of equation (4), we have $\\{(j, k) \in E: \mathrm{dist}(i, j) < \ell, \mathrm{dist}(i, k) \le \ell \\}$. A better writing should be $\\{(j, k) \in E: \mathrm{dist}(i, j) \le \ell, \mathrm{dist}(i, k) < \ell \\}$, which is consistent with our usage of $j$ in the node set. It means that: if $(j, k)$ is included in the edge set of $S_i^\ell$, then it must in a length-$\ell$ walk starting from $i$, then the distance between $i$ and one end needs to be less than $\ell$.
>
> --- Page 4: For the function id(·) that maps a tree node to a node in an anchored graph, since a node in an anchored graph may appear multiple times in the tree, is it still a function?
>
> It is a function since it may map multiple tree nodes (multiple appearances of a graph node) to a graph node.
>
> --- [W2] Consider a counter-example, where G is a graph consisting of two triangles and H is a cycle of length 6. These two graphs cannot be distinguished by 1-WL, but would have different WL-trees proposed in the paper.
>
> We will have the same WL tree, which has the following pattern: each node has two children.
>
> --- [W3] It is unclear why the proposed WL-trees can provide a more fine-level analysis of the expressiveness of node representations learned by message-passing GNNs. In particular, the proposed WL-trees are not equivalent to 1-WL (see the above point [2]).
>
> As we have shown in our work, a WL-tree is equivalent to a WL-color. We have clarified W2 as well. In this regard, our WL-trees provide a new way of analyzing graph structures behind node representations. It is easier to use because 1) it connects well with the original graph structure, and 2) it is easy to visualize.
>
> --- [W4] In what kinds of scenarios will the proposed algorithm 1 be useful?
>
> The algorithm is able to enumerate graph structures that give the same WL tree/color. When we want to analyze these structures, e.g. checking which structures a GNN cannot distinguish, then we can use the algorithm to do so. In fact, we use the algorithm to check how many graph structures can possibly be confused by GNNs in section 6.
>
> --- [W5] For the section 6, what are the justifications for selecting CLIP and Nested GNN? There are a large number of GNN models developed in the literature. I don't see why these two particular GNN models are selected for analysis.
>
> Because these two GNNs can be best explained by our WL-trees. We plan to examine the expressiveness of more GNNs in our future work.
>
> --- [W6] Theorem 12 and Theorem 13 look confusing. Why is max used in Equation 8? Is the notation
>  defined?
>
> The max operation is taken over all possible colorings $\bar{c}$. It is from the CLIP-2 method -- we just formalize the method.
>
> --- Also, the expressive power of Nested GNN goes beyond 1-WL, but the WL-trees proposed in the paper are claimed to be equivalent to 1-WL. So why does Theorem 13 state that there is a bijective mapping between WL-trees and their node embeddings calculated by Equation 10?
>
> Both the two GNNs first color graph nodes with extra information (random colors or local message-passing) and then run a vanilla GNN to get node representations. So we can use our WL trees to quantify the abilities of their vanilla GNNs using extra node colors.
>
> --- [W7] For the statement "A smaller count or conditional entropy means that the WL-tree can better identify a node’s surround structure", is any theoretical justification? Analysing GNN models using average counts of anchored subgraphs and the conditional entropy of anchored graphs look ad hoc.
>
> The conditional entropy is computed from possible graph structures in the dataset that give the same WL-tree/color. If the conditional entropy is 0, it means that there is only one graph structure giving the WL tree/color. In our view, the analysis with an information quantity is reasonably because it tights to a GNN's ability in terms of distinguishing graph structures.

---

### Official Review · Reviewer_wEKA · 2023-10-27

**Soundness:** 2 fair
**Presentation:** 2 fair
**Contribution:** 2 fair
**Rating:** 3
**Confidence:** 3

**Summary:**

This paper proposes a new concept, WL-tree, as a new perspective for analyzing GNNs. It theoretically proves that WL-trees are bijective mapping with the colors given by the 1-WL algorithm. It claims that such a new perspective could bring new understandings of the encoded structural information in GNN node representations.

**Strengths:**

1. The motivation of analyzing what structural informative is encoded in node representations is important.

2. The formulations of the theorems in the paper are formal, which could be potentially useful for the community.

**Weaknesses:**

1. Although WL-tree seems to be a new concept, I did not quite get what perspective from which it is important compared to existing 1-WL algorithm results. It is known that message passing GNN, such as GIN, are capturing the rooted subtree around each node, which is exactly the structure captured by 1-WL, as shown in Figure 1 of [1]. As defined in Section 4, the WL-tree proposed in this paper is also such a rooted subtree. The only difference is that the rooted tree in this paper does not include the parent of a node as its child. It is not clear to me why this difference is important and how it brings significant differences compared to existing understanding. (Please correct me if I am understanding wrongly or incompletely.)

2. Also, it is unclear what advantages or new understandings can be inspired by this new concept. Could you summarize what findings we can get from this new analysis tool?

3. The experiments only show a simple analysis of two existing GNN models. What new model designs this new concept can lead to? This is not obvious from the reading. I think the experimental section can include deeper analyses or include a model inspired by the introduced WL-tree tool.




[1] Xu, Keyulu, et al. "How Powerful are Graph Neural Networks?." International Conference on Learning Representations. 2018.

**Questions:**

See above

---

> ### Author Response · Authors · 2023-11-23
> **Thank you for your comments**
>
> Thank you for seeing our effort in a formal analysis of GNN's expressive power. Below are answers to your questions.
>
> 1. "It is not clear to me why this difference is important and how it brings significant differences compared to existing understanding."
>
> There are at least two benefits if we do not include the parent in the child. First, it is easier to discuss a node's WL color in a tree -- our formulation directly uses the original tree structure without adding new nodes. Second, it is easier to enumerate graph structures behind a WL color. With our formulation, we only need to merge some nodes to get a graph structure. If a node's parent is also included in the child list, then the situation is much more complex.
>
> 2. " Could you summarize what findings we can get from this new analysis tool?"
>
> First, with the new tool, it is easier to discuss which graph nodes have the same color/node representations. For example, showing that a GNN with a fixed depth cannot distinguish a graph node from a particular tree node. Second, it is easier to explain how previous GNNs enhance node representations. Random node colors give a particular tree pattern (some nodes much take the same color), so that the graph structure can be separated from others that have the same WL color.
>
> As we mentioned in our submission, this new tool is fundamentally no different from WL colors but it makes some analysis clearer, particularly when we can visualize examples with WL trees. As a comparison, it is more abstract to discuss WL colors and logic to reach the same conclusion.
>
>
> 3. Findings from our experiments
>
> In our experiment, we use our new method to enumerate graph structures corresponding to a WL tree/color. We show that advanced GNN methods are able to eliminate a large number of possible graph structures by including extra node colors. But they do not do much better in terms of distinguishing graph structures **in the dataset**: a WL tree/color may correspond to many graph structures, but if only one of these structures is in the dataset, then it still can be distinguished by a vanilla GNN. Our experiment explains why we don't see a significant increase in performance when we use stronger GNNs on some datasets.

---

### Meta-Review · Area_Chair_U6cx · 2023-12-03

**Metareview:**

This work introduces a tool for analyzing Graph Neural Networks beyond the WL-test and proposes WL-Trees, a new algorithmic model of GNNs. However, the authors fail to differentiate their new methods from the rich literature on graph modeling, and it is unclear what advantages or new understandings/networks can be inspired by this new concept. Given that all reviewers felt this work didn't meet the acceptance bar, I recommend rejection.

**Justification For Why Not Higher Score:**

See the metareview.

**Justification For Why Not Lower Score:**

N/A

---

### Decision · Program_Chairs · 2024-01-16

Reject